# Effect of Thermal and Hydrothermal Accelerated Aging on 3D Printed Polylactic Acid

**DOI:** 10.3390/polym14235256

**Published:** 2022-12-01

**Authors:** Saltanat Bergaliyeva, David L. Sales, Francisco J. Delgado, Saltanat Bolegenova, Sergio I. Molina

**Affiliations:** 1Department of Materials Science and Metallurgical Engineering and Inorganic Chemistry, Algeciras School of Engineering and Technology, Universidad de Cádiz, INNANOMAT, IMEYMAT, Ramón Puyol Ave, 11202 Algeciras, Cádiz, Spain; 2Physics and Technology Department, Al-Farabi Kazakh National University, 71, Al-Farabi Ave, Almaty 050040, Kazakhstan; 3Department of Materials Science and Metallurgical Engineering and Inorganic Chemistry, Universidad de Cádiz, Campus Río S. Pedro, INNANOMAT, IMEYMAT, 11510 Puerto Real, Cádiz, Spain

**Keywords:** polylactic acid, thermal aging, hydrothermal aging, 3D printing, additive manufacturing, fused filament fabrication, recyclability

## Abstract

In the new transformation of ‘Industry 4.0’, additive manufacturing technologies have become one of the fastest developed industries, with polylactic acid (PLA) playing a significant role. However, there is an increasing amount of garbage generated during the printing process and after prototypes or end-of-life parts. Re-3D printing is one way to recycle PLA waste from fused filament fabrication. To do this process successfully, the properties of the waste mixture should be known. Previous studies have found that PLA degrades hydrolytically, but the time at which this process occurs for 3D printed products is not specified. This work aims to establish the baseline of the degradation kinetics of 3D printed PLA products to predict the service time until which these properties are retained. To achieve this, 3D printed specimens were thermally and hydrothermally aged during several time intervals. Thermal and mechanical properties were also determined. This study reveals that tensile strength decreases after 1344 h of hydrothermal ageing, simulating 1.5–2.5 years of real service time. PLA therefore has the same thermo-mechanical properties before reaching 1.5-years of age, so it could be recycled.

## 1. Introduction

Some industrial leaders and experts predict that, by 2030, two thirds of all manufactured products in the world will be produced with 3D printed components [1]. Compared to traditional production methods, additive manufacturing (AM) is quickly and inexpensively set up closer to the point of consumption, thus simplifying logistics and transportation, which is why it is globally used. Moreover, this technology reduces carbon footprint. From 2019 to 2050, the cumulative greenhouse gas emissions reduction of lightweight metallic aircraft components produced using AM is estimated at 92.8–217.4 Mt [2].

Fused filament fabrication (FFF) is the most used AM process because of its simplicity, low cost of running, and material costs [3]. Polylactic acid (PLA) is among the most popular polymers for 3D printing. Despite its relatively high tensile strength and modulus compared to other thermoplastics (e.g., polyethylene terephthalate and polypropylene), its low impact strength, less heat tolerance, brittleness with less than 10% elongation at break [4], low crystallization rate, and poor ductility make it inappropriate for more demanding applications [4,5]. However, PLA has a relatively low melting point (150–160 °C) and is a safer alternative to acrylonitrile butadiene styrene, another popular AM polymer [6].

PLA is a biodegradable and renewable thermoplastic polyester from renewable sources (mainly starch and sugar) that could replace conventional petrochemical-based polymers and reduce oil consumption by 30–50% [4,5,7,8]. However, Kolstad et al. [9] estimated that biodegradation in landfills at 20 °C will take 100+ years. Microorganisms are required to be decomposed in natural environments [5,10,11]. This slow degradation rate could lead to PLA accumulation in the environment [4,12,13,14,15]. In addition, PLA production cost is currently high, compared to conventional petroleum-derived plastic products [16]. Effective recycling methods for PLA waste should therefore be developed. Furthermore, research on recycling plastics from 3D printing is limited [17]. Some works have tried to solve this problem. For example, Baechler et al. [18] developed an extruder to produce filament from polymer waste; Filabot [19] designed a plastic filament maker that uses post-consumer plastic; Woern et al. [20] designed an open source recycling system that costs less than USD 700 in materials and can be fabricated in about 24 h; and Dongoh Lee et al. [21] designed a recycling system to make post-consumer filaments for 3D printers.

The degradation processes that polymeric materials are exposed to in each step of their life cycle is a fundamental factor to consider when discussing both their further waste recovery possibilities and the performance of recycled plastics [22]. Previous studies have reported that PLA degrades during thermal processing or under hydrolytic conditions, rapidly reducting the molecular weight, which affects the final properties of the material, such as mechanical strength [23]. The results of [24] indicated that hydrothermal aging of injection molding grade PLA, corresponding to 1 year of service at room temperature, significantly degraded PLA, thus reducing glass transition, cold crystallization, and melting temperatures. Mechanical properties of aged samples could not be measured. Moreover, compressing molded PLA bars after accelerated hydrothermal aging for 30 and 60 days significantly reduced the molecular mass of materials [25].

Both accelerated thermal and hydrothermal aging are reliable ways to obtain aged samples. There are standardized methods for thermal aging. For example, ISO 188 [26] describes a testing methodology for all types of plastics. However, the specifics of the materials should be considered. A normalized water absorption test reported in the Method 1 ISO 62 [27] can be adopted for hydrothermal aging, as in Refs. [28,29,30]. This method involves the complete immersion of the sample in water for a certain time and at a certain temperature. However, the specificity of FFF products is both an increased roughness and the presence of pores between the layers. Water therefore enters through these holes, and sample breaks down faster. Most studies have reported that hydrolytic degradation of PLA takes place in the bulk of the material rather than at the surface [31]. This study proposes a hydrothermal aging of PLA specimens under real conditions of indoor operation of PLA 3D printed samples. As a result, data were used both to study the effect of thermal and hydrothermal aging on FFF PLA in detail and to predict the service time of the FFF PLA printed part to assess its recyclability.

This work presents the results of dimensional, thermal, and mechanical properties of 3D printed PLA specimens after 8, 16, 24, 48, 72, 168, 672, and 1344 h of thermal aging in a laboratory oven, as well as after 24, 48, 72, 168, 672, and 1344 h of hydrothermal aging in a climatic chamber. These data showed the changes in the structure of PLA specimens produced by FFF technology during accelerated treatment. Moreover, the recyclability of the PLA polymer used in AM is predicted, and the printing parameters that should be used for successfully re-manufacturing recycled PLA filament were shown. The novelty of this study is based on the use of 3D printed PLA samples for testing instead of injection molding, the setting of the aging temperature below $T_{g}$ (glass transition temperature) to avoid the interphase transition, and the consideration of hydrothermal aging with controlled ambient humidity instead of immersing samples in water.

## 2. Materials and Methods

### 2.1. Materials

Commercially available PLA filament of 1.75 mm width was purchased from BQ (Madrid, Spain), with a printing temperature range = 200–220 °C, an optimum printing temperature = 205 °C, a bending temperature under load = 56 °C (ISO 75/2B), a melting temperature = 145–160 °C (ASTM D3418), and a glass transition temperature = 56–64 °C (ASTM D3418), as indicated in the product datasheet.

### 2.2. Printing

A total of 90 regular dog-bone (type A2) specimens for thermal and hydrothermal aging were printed in batches of 5 considering the ISO 20753 standard [32] and using a Witbox 2 printer (BQ) with operating temperatures of 200 °C and a printing glass table without heater. Five specimens are used for each aging cycle, and one extra specimen for contingency. All specimens were printed indoors in a temperature-controlled environment (23±2 °C and 50±5% R.H., as defined in Ref. [27]) with 100% infill, horizontal pattern orientation 0/90 (i.e., alternating layers with orientations at 0° and 90°), and a layer height of 0.2 mm. Each specimen was printed with a skirt of two extruded perimeters that composed the outermost edge. The selected deposition pattern orientations and layer height offered better quasi-isotropic mechanical properties for PLA parts manufactured by FFF compared with 45/45 orientation [33,34]. Likewise, the Ultimaker Cura 4.1.0. (Ultimaker B.V.) software was used for slicing the STL files into machine-readable G-code.

### 2.3. Thermal Aging

Accelerated thermal aging of 3D printed specimens from PLA filament was carried out in a laboratory Selecta (Spain) oven at 50±2 °C. This aging temperature was chosen to be under the glass transition temperature ($T_{g}$) of PLA to prevent interphase transition [35]. Samples were removed at different time intervals (8, 16, 24, 48, 72, 168, 672, and 1344 h) as indicated in ISO 188 [26]. The simplified protocol for accelerated aging states that service time of one month at room temperature can be simulated by an aging time of 72 h at 50 °C with thermo-oxidative degradation [24,36]. According to this protocol, Table 1 includes the correlation between aging and service time for the intervals selected for this experiment. The maximum thermal aging time in this experiment is 1344 h, corresponding to about 1.5–2.5 years of real service time.

Temperature stability was controlled during aging at three points by electronic thermocouples Testo (Germany), with a constant temperature recording with intervals of 30 s. The points were as follows: in the laboratory air, on the surface of one of the specimens, and in the central point of the oven air without touching any surfaces.

### 2.4. Hydrothermal Aging

There is no international standard that regulates the conditions of hydrothermal aging tests for plastic materials. After studying the most important works on the hydrothermal aging of PLA [24,25,37,38,39,40,41,42,43], the test was conducted in a climatic chamber, at a temperature of 50±2 °C and humidity of 70±5%. Aging for 8 and 16 h corresponds to a short use period in real life, i.e., only 1–2 weeks, so these time intervals are not included in the hydrothermal aging experiment. Specimens were aged 24, 48, 72, 168, 672, and 1344 h.

### 2.5. Characterization of Dimensional Stability after Aging Conditions

Thickness, width and length of each specimen were measured three times before and after aging with a micrometre, as per ISO 16012—Plastics—Determination of linear dimensions of test specimens [44]. Specimens were conditioned for at least 88 h at 23±2 °C and 50±5% R.H. atmosphere, as defined in ISO 291—Plastics—Standard atmospheres for conditioning and testing [27] with ±0.02 mm accuracy.

### 2.6. Characterization of Thermal Properties

A Q20 differential scanning calorimeter (DSC) from TA Instruments Inc. (USA) was used to measure the thermal properties of potential PLA waste, as indicated in ISO 11357-1 [45]. Measurements were carried out using nitrogen purge gas of purity 99.99% to avoid oxidative or hydrolytic degradation while testing at 10 mL/min flow rate with heating run at a rate of 10 °C/min. Specimens were cut with a razor blade from 3D printed aged pieces, from 5 mg to 10 mg for measurements. Moreover, they were weighed with aluminum crucibles with an oxidized surface and their lids with a resolution of ±0.01 mg and an accuracy of ±0.1 mg. Data were obtained in the first heating cycle to evaluate the properties of especially pre-conditioned specimens (aged for 8, 16, 24, 48, 72, 168, 672, and 1334 h) without erasing the material’s previous history. All data are reported to the nearest 1 °C, as per ISO 11357-1.

### 2.7. Characterization of Thermal Stability

Additionally, a thermogravimetric analysis (TGA) was performed to identify the thermal stability of PLA samples using the Thermogravimetric analyzer Q50 from TA instruments Inc. (USA). Approximately 15 mg of polymer were heated from 30 to 430 °C at 10 °C/min, under N${}_{2}$ flow at 10 mL/min. A temperature of 5% wt. mass loss $\left(T_{loss5\%}\right)$ was determined.

### 2.8. Characterization of Mechanical Properties

Finally, the mechanical properties of aged PLA test specimens were determined through tensile testing. All tests were carried out after dimensional measurements, so the specimens were already conditioned (see Section 2.5). Uniaxial tensile testing was performed on the specimens along its main longitudinal axis at a constant speed of 2 mm/min until failure and using a universal testing machine Shimadzu AG-X series (Japan) in a standard atmosphere (temperature was 23±2 °C and relative humidity 50±5%) as per ISO 527-2 [46]. The load sustained by specimens and elongation were also measured. An extensometer with a nominal length of 50 mm was used, but many specimens broke outside the gauge length due to assumed stress concentrations in the regions where geometry changes as in Ref. [34]. Data of specimens that broke outside the gage length but displayed a distinct maximum stress before failure were also included.

Young’s modulus was calculated as the slope between 0.05% and 0.25% strain on a stress-strain plot. Yield strength was obtained as the stress value where an increase in strain does not increase stress (maximum value of Y-axis). Elongation at break was obtained as the strain value in the rupture point (maximum value of X-axis). Results were averaged and standard deviations were presented as error bars. The results of tensile tests for 3D printed materials were transmitted using the Trapezium software. Material properties were calculated using a spreadsheet (Table 3).The repeatability of the individual measurements is calculated as the difference between the maximum and minimum values among five samples.

## 3. Results and Discussion

### 3.1. Dimensional Stability

Figure 1 shows the results of three perpendicular dimensions of specimens before and after thermal and hydrothermal aging. Shrinkage ratio of the length is calculated using the following equation:
(1)$$R=\frac{100\cdot (L_{0}-L_{a})}{L_{0}},$$
where *R* is the shrinkage ratio, $L_{0}$ is the whole length of the specimen before aging, and $L_{a}$ is the whole length of the aged specimens [47]. The shrinkage ratio of width and thickness are calculated in the same way.

The results of the dimensional measurements show that PLA molecules moved during aging. This molecular rearrangement is mentioned in Refs. [47,48]. According to data in Figure 1, the shrinkage ratio in width direction for most thermal aging periods, except 72 and 168 h, does not exceed 1%. On the other hand, the shrinkage ratio in the width direction for hydrothermally aged specimensis approximately the same in the range between 1 and 2% until 168 h, and it constantly grows after 672 h, reaching 3.14% after 1344 h of hydrothermal treatment. Water and temperature significantly influenced PLA molecular mobility.

Shrinkage ratios in length and thickness directions, as shown in Figure 1, obtain the same values with opposite signs after each aging period. Moreover, values of shrinkage ratios of length and thickness directions of the specimens influenced only by temperature are lower than those treated by temperature and humidity. The values of shrinkage ratios of thermal aged PLA had no constant tendency; however, there is a rising tendency in hydrothermal samples. Finally, it can be supposed that printing the skirt in FFF does not let the width change. Although shrinkage is not strong during aging, it should be considered when designing goods manufactured through FFF where high accuracy is required.

Accelerated aging for 8 h, which simulates using 3D printed PLA goods for 2.5–5 days under real service conditions, shows that a short use period changes geometric dimensions. As shown in Ref. [49], the physical aging rate is first fast and then it decreases as time increases. Geometric dimensions of any PLA product manufactured through FFF would therefore change in the first days after printing. Afterwards sizes would stay the same for 1–2 years, with a maximum shrinkage ratio of 8% after water and temperature influences.

### 3.2. Calorimetry

Figure 2 shows the DSC results of unaged and aged PLA samples. The first shift of the curves down corresponds to $T_{g}$. For unaged polymer $T_{g}=58$ °C. The next exothermic peak is due to a crystallization process ($T_{c}$, temperature of crystallization) in which the polymer gives off heat. Unaged PLA reaches this point at 117 °C. The third peak shows that the polymer achieves its melting point ($T_{m}=151$ °C for unaged PLA). Figure 2 shows that all curves correspond to semicrystalline PLA, showing both glass transition and melting peaks [50]. Likewise, $T_{g}$, $T_{c}$, and $T_{m}$ of unaged PLA specimens are marked with a vertical thin continuous line to easily compare the unaged sample and each aged one.

As summarized in Table 2 (third column), $T_{g}$ slightly fluctuates along the aging within a range of 4 °C (between 58 and 62 °C), with no constant tendency. Nevertheless, the results show that the $T_{g}$ of aged PLA is equal or greater than that measured for the unaged sample, so the aged material requires greater temperature to become rubber-like. In addition, $T_{g}$ influences the extrusion process, the shrinking of parts during the cooling process, and the thermostability of the final part [51]. Based on the value of $T_{g}$ and its influence on the 3D printing process it can be suggested that the re-extrusion of PLA has no shrinkage problems during re-printing. Additionally, Table 2 (fourth column) shows that the $T_{c}$ of PLA hydrothermally aged during 1344 h decreases. The value of $T_{c}$ drops by 14 °C in relation to the unaged sample. The reduction in $T_{c}$ can be attributed to the higher mobility of the polymer chains, as a consequence of the reduced molecular weight [52].

Table 2 shows that $T_{m}$ fluctuates from 149 to 155 °C. Both the rising and decreasing tendency in $T_{m}$ during thermal treatment could be due to molecular rearrangements with aging [7]. The mobility of PLA molecules is shown by the dimensional measurements of specimens before and after aging. Shorter polymer chains effectively reorganize themselves into more ordered crystals, thus increasing the relevance of the high-temperature melting peak [52].

DSC thermograms of thermal and hydrothermal aging have two melting peaks after 672 and 1344 h of treatment. The reason could be the annealing during the DSC scans, where some regions of the material recrystallize and remelt [8]. When the scan rate is low, i.e., 10 °C/min, there is enough time for thinner crystals to melt and recrystallize before a second endotherm at a greater temperature occurs [53]. The second peak (the one at greater temperature) was therefore considered as the melting point for samples with a double peak and is registered in Table 2 in column $T_{m}$. According to the received data, enthalpies of crystallization exceed the enthalpies of melting for thermally aged samples during 16, 24, 48, and 72 h. The relation $\Delta H_{c}\approx \Delta H_{m}$ indicates that almost the entire crystal phase in PLA forms during the cold crystallisation, which excludes the crystallisation from the liquid phase [54]. As the printed samples were cooled down at ambient temperature it resulted in the quenching of the molten PLA [55]. The inequality $\Delta H_{c}>\Delta H_{m}$, observed for thermally aged samples during 16, 24, 48, and 72 h, may be the result of an indistinct separation of the processes of crystallisation and melting, which leads to a partial superposition of the relevant peaks. The same phenomenon was observed by Ref. [54].

The degree of crystallinity $X_{c}$ was quantified according to Refs. [52,54,55,56] as
(2)$$X_{c}=\frac{\Delta H_{m}-\Delta H_{c}}{\Delta H*}\cdot 100$$
where ΔH∗=93 J/g denotes the heat of melting for an infinitely large crystal [7]. $X_{c}$ for thermally aged during 16, 24, 48, and 72 h are not considered because of the previously mentioned partial superposition of crystallization and melting peaks. The degree of crystallinity for thermally aged specimens stays constant at about 0–2.2%. However, for hydrothermally aged PLA, the degree of crystallinity is rising from 1.1% for unaged PLA to 8.6% after 1344 h. It can be because linear macromolecules do not normally crystallize completely; they are semicrystalline, as mentioned previously. The restriction to crystallization is caused by a kinetic hindrance to the full extension of the molecular chains which, in the amorphous phase, are randomly coiled and entangled [57]. As crystallinity is an important characteristic affecting the mechanical properties [58], its influence on mechanical properties will be shown in Section 3.4.

So, for thermally aged PLA, $T_{g}$, $T_{c}$, $T_{m}$, and $X_{c}$ do not change much; for hydrothermally aged, $T_{g}$ and $T_{m}$ stay the same, but $T_{c}$ is decreasing and $X_{c}$ is increasing. It becomes apparent, therefore, that the degradation is mainly caused by a hydrolytic process, rather than a thermal one.

### 3.3. Thermogravimetry

Thermal decomposition properties of PLA were studied by TGA from room temperature to 430 °C in nitrogen. Figure 3 shows the parts of TGA curves from 230 to 400 °C for all specimens. Unaged PLA is very stable up to 300 °C. The mass hardly changed from room temperature to 300 °C, so the samples have no contaminates, are dry with no solvent, and stable [50]. Additionally, all samples show one single weight reduction step, indicating the presence of one type of polymer (PLA).

The values of $T_{loss5\%}$ (last column in Table 2) indicate that the thermal stability of aged PLA does not constantly increase or decrease. Despite the absence of a constant tendency to rise or fall, a reduction of $T_{loss5\%}$ of thermally and hydrothermally aged samples during 1344 h can be observed compared with the $T_{loss5\%}$ of the unaged sample. Consequently, 3D printed PLA goods had no predictable thermal behavior during their life cycle because of the unstable thermal nature of PLA.

### 3.4. Tensile Testing

The results of tensile testing with 95% confidence intervals of values, according to the procedure given in ISO 2602 [59], are shown in Table 3 and Figure 4. Tensile strength gradually increases up to 24 h of aging, and then decreases to both 31.126 MPa for thermal aged and 20.890 MPa for hydrothermally aged specimens after 1344 h.

The average tensile strength for the thermally aged specimens after 1344 h of aging is 31.126 MPa. An analysis of variance (ANOVA) has been performed using StatGraphics software (Statgraphics Technologies, Inc., The Plains, VA, USA) to check whether the differences observed among the tensile strength mean values are statistically significant. Since the calculated *p*-value of the ANOVA F-test is zero (less than 0.05), there is a statistically significant difference between the yield strength from one thermal aging time to another at a confidence level of 95.0%. Therefore, it can be stated that, for thermally aged samples, the tensile strength rises during the first 24 h. Then, after 48 h, the tensile strength reduces back to values similar to those of the unaged specimens and does not show significant changes after further aging. It is important to note that all specimens for tensile analysis have been tested after determining the geometric sizes, and, following ISO 291, they were conditioned for 88 h with standard temperature and relative humidity. PLA is hygroscopic, and specimens could reabsorb water molecules from ambient humidity after 3D printing, which are weakly bound to the carbonyl oxygen atom of PLA through dipole interactions. Water may subsequently be evaporated in the first 24 h of thermal aging [31], so the PLA chains are able to create more Van Der Waals interactions between them, thus improving tensile strength. The reduction in strength observed after 48 h of thermal aging is consistent with the slight decrease in $T_{m}$ and $T_{c}$ mentioned in Section 3.2, which suggests a chain length shortening.

The tensile strength drops to 20.890 MPa after 1344 h in the climatic chamber. This fact occurs when the crystallinity doubles from 672 h to 1344 h of hydrothermal aging. This weakening effect could be due to the anisotropy of the crystalline part of the polymer. Thus, it can be concluded that, when crystallinity is low, the tensile strength does not change.

The degradation of 3D printed PLA after hydrothermal aging was somewhat less dramatic to that of injection molded PLA, as reported by Yarahmadi et al. [24] under similar hydrothermal conditions. Aging corresponding to one year of use significantly degraded PLA, so the mechanical properties could not be measured. In addition, PLA aging significantly decreased $T_{g}$, $T_{c}$, and $T_{m}$.

## 4. Conclusions

Lab-controlled aging of PLA has been considered to predict their long-term behavior for future recycling. The results of dimensional measurements, DSC, TGA, and tensile tests have shown the degradation process in thermal and hydrothermal aging up to 1334 h (i.e., 1.5–2.5 years of real service life cycle).

After aging, the PLA parts experience a shrinkage ratio from 1 to 8%, according to the printing pattern (actual material density in the measured dimension), aging time, and nature (thermal or hydrothermal). The shrinkage ratio for hydrothermal aged specimens is greater. The calorimetry test shows that $T_{g}$ and $T_{m}$ is the same for both types of aging, with a maximum shift of 6 °C. The reason could be molecular rearrangements, also observed in dimensional measurements. The degree of crystallinity for hydrothermal aged PLA continuously rises up to 8.6%. The results of the TGA test confirm the unstable thermal nature of PLA. The TGA curves of the aged samples change their shape from sharp drop curves to slopes with lower angles. Moreover, the mechanical properties of PLA show stable conditions. However, the tensile strength decreases to 20.890 MPa (33% reduction) after 1344 h of hydrothermal treatment.

After comparing the results of this study with those in previous studies on molded PLA, it can be concluded that FFF processed PLA had slower degradation kinetics.

According to data, PLA waste before 1.5 years of age could be mechanically recycled without risk of substantial loss of properties, compared to unaged material. It is also worth noting that PLA waste mixture has the same thermo-mechanical properties before reaching 1.5 years of age, so it could be mixed for recycling. Finally, future dimensional changes should be considered when designing and printing goods through FFF.

## Figures and Tables

**Figure 1 polymers-14-05256-f001:**
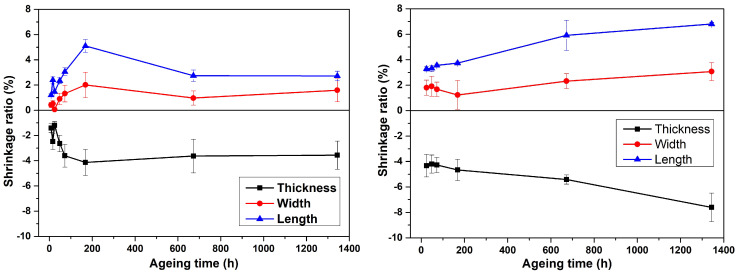
Shrinkage ratios in width, length, and thickness directions versus aging time for thermal aging (**left**) and hydrothermal aging (**right**). Error bars are inserted considering standard deviations for six calculated shrinkage ratios.

**Figure 2 polymers-14-05256-f002:**
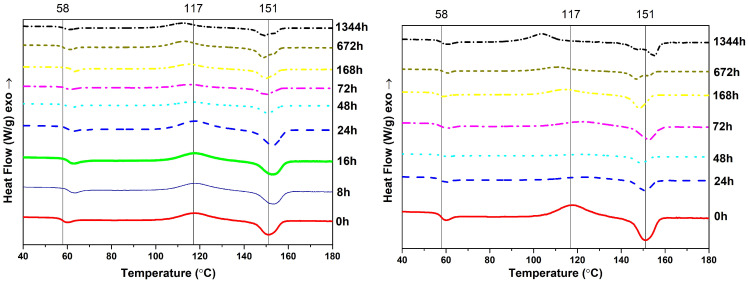
DSC test of PLA specimens after thermal aging (**left**) and hydrothermal aging (**right**). The lines at 58, 117, and 151 °C correspond to $T_{g}$, $T_{c}$, and $T_{m}$ of unaged PLA specimens, respectively. The graphs are shifted in the Y-axis to better compare the peak temperatures.

**Figure 3 polymers-14-05256-f003:**
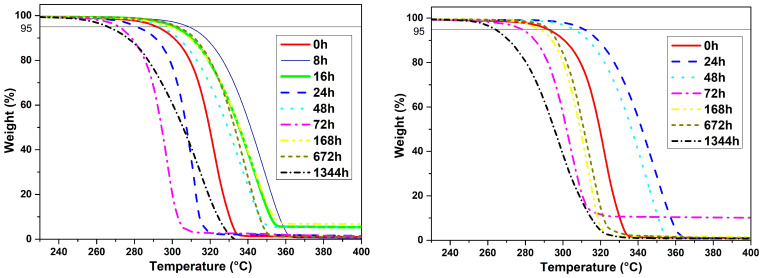
TGA curves of PLA specimens after thermal aging (**left**) and hydrothermal aging (**right**).

**Figure 4 polymers-14-05256-f004:**
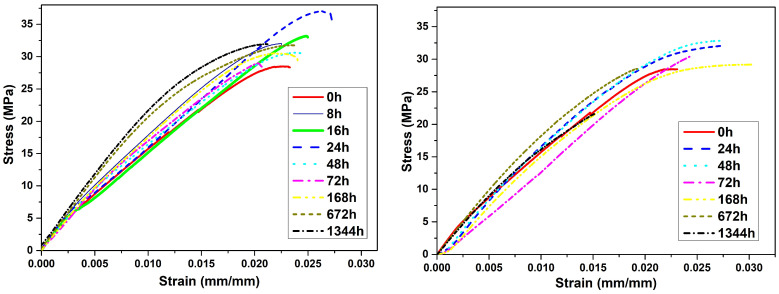
Stress–strain curves after thermal aging (**left**) and hydrothermal aging (**right**).

**Table 1 polymers-14-05256-t001:** Correlation between aging and service time as per the simplified protocol for accelerated aging.

Aging Time at 50±2 °C	Approximate Time
(h)	(Day)
0	0
8	2.5–5
16	5–10
24	10–16
48	16–24
72	24–56
168	56–112
672	224–448
1344	448–896

**Table 2 polymers-14-05256-t002:** Results of the thermal analysis of the specimens after aging. Samples that registered a double peak when melting are marked with an asterisk (*). The second peak (the one at a greater temperature) is also shown, considered as $T_{m}$.

	Aging Time	$T_{g}$	$T_{c}$	$\Delta H_{c}$	$T_{m}$	$\Delta H_{m}$	$X_{c}$	$T_{\mathrm{loss}5\%}$
	(h)	(${}^{\circ }$C)	(${}^{\circ }$C)	J/g	(${}^{\circ }$C)	(J/g)	(%)	(${}^{\circ }$C)
Unaged	0	58	118	27	151	28	1.1	292
Thermal	8	61	118	27	153	28	1.1	309
	16	61	118	30	153	28	-	301
	24	60	118	29	153	28	-	281
	48	59	117	32	149	27	-	294
	72	60	115	32	149	29	-	272
	168	62	115	26	150	28	2.2	298
	672	59	113	29	149 *	29	0	303
	1344	59	112	27	149 *	28	1.1	266
Hydrothermal	24	58	125	15	151	18	3.2	311
	48	59	123	11	149	20	4.3	305
	72	58	123	16	153	20	4.3	278
	168	57	115	23	149	27	4.3	287
	672	59	111	27	147 *	31	4.3	303
	1344	58	104	31	155 *	39	8.6	264

**Table 3 polymers-14-05256-t003:** Tensile properties of the 3D printed specimens after aging.

	Aging Time	Tensile Yield Strength			Young’s Modulus			Elongation		
		Average	Standard Deviation	Repeat-Ability	Average	Standard Deviation	Repeat-Ability	Average	Standard Deviation	Repeat-Ability
	(h)	(MPa)	(MPa)	(MPa)	(MPa)	(MPa)	(MPa)	(%)	(%)	(%)
Unaged	0	28.746	1.002	2.226	1800.208	150.395	323.839	2.34	0.19	0.45
Thermal	8	31.189	1.322	3.378	2158.489	117.927	307.523	2.19	0.14	0.33
	16	33.164	1.129	2.775	2115.412	157.063	356.287	2.39	0.16	0.37
	24	36.799	1.154	2.894	2134.546	286.843	475.852	2.53	0.21	0.50
	48	29.135	1.681	3.845	1612.385	312.078	429.898	2.49	0.22	0.54
	72	29.721	1.551	3.875	1875.389	71.578	152.065	2.28	0.21	0.56
	168	29.516	0.495	1.133	1968.879	102.064	261.783	2.05	0.23	0.57
	672	32.602	0.477	0.927	2141.455	48.957	123.278	2.06	0.46	0.98
	1344	31.126	2.851	7.579	2077.937	186.864	443.046	2.00	0.46	0.66
Hydrothermal	24	33.454	1.238	3.328	1552.181	74.724	177.765	3.16	0.15	1.30
	48	32.672	2.040	4.950	1652.115	243.715	651.275	2.94	0.43	1.06
	72	32.530	1.218	2.943	1340.578	322.173	825.923	3.05	0.29	0.60
	168	29.613	1.259	2.441	1664.317	144.847	365.233	2.81	0.49	1.20
	672	29.816	2.190	4.598	1962.120	94.488	237.910	2.40	0.57	1.35
	1344	20.890	4.240	10.727	1598.108	383.587	936.971	1.74	0.19	0.46

## Data Availability

Data are available upon reasonable request by contacting the corresponding author.

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
