# Peer review of "Effect of Thermal and Hydrothermal Accelerated Aging on 3D Printed Polylactic Acid"

_polymers, 2022, doi:10.3390/polym14235256_

Round 1
Reviewer 1 Report
The article discusses a study on ageing of 3D printed PLA parts - both thermal and hydrothermal. The aged parts have been subjected to mechanical and thermal test to quantify the degradation in properties. The experimental setup and conditions are discussed well and the results are presented in the forms of graphs/tables. However, some revisions are needed to improve the manuscript:
1. First and foremost, there needs to some discussion on how the degradation kinetics of 3D printed PLA differ from that of injection molded PLA or other common techniques.
2. The Introduction needs to be written in a manner that the flow is clear. As of now, it appears that the sentences are disjointed.
3. Missing commas in multiple places. For example, Page1, Line 18. by 2030,
4. Page 2, Line 44: Incomplete sentence
5. Page 2, Line 63-64: Reword sentence.
6. Page 3, Line 91: provide correct print temperature. Also indicate that a skirt was used. Were all 100 samples printed in one go or in batches?
7. Page 3, Line 109: It is not clear as to what the three points are.
8. Page 4: Line 122-123. Multiple ISO standards are referenced in the manuscript without indicating what they stand for. I can be stated as for example, atmosphere defined in ISO 291 - Plastics - Standard atmospheres for conditioning and testing.
9. Page 4: Line 152 - Reword sentence. Not clear.
10. Page 5: Line 178 - Table 2 and Fig. 1 convey same information. Use one of them.
11. Page 5: Line 183: Numeral?
12. Page 5, Line 194: Reword sentence.
13. Page 7, Line 231: add Section before 3.4
14. Page 8, Line 245: have not must be have no
Author Response
Point 1: First and foremost, there needs to some discussion on how the degradation kinetics of 3D printed PLA differ from that of injection molded PLA or other common techniques.
Response 1: A paragraph has been included accordingly in the manuscript at the end of the Introduction, Line 54-59:
“The results of [24] indicated that hydrothermal ageing of injection molding grade PLA corresponding to 1 year of service at room temperature significantly degraded PLA, thus reducing glass transition, cold crystallization, and melting temperatures. Mechanical properties of aged samples could not be measured. Moreover, compressing molded PLA bars after accelerated hydrothermal ageing for 30 and 60 days significantly reduced the molecular mass of materials [25]”
Results, Line 294-298:
“The degradation of 3D printed PLA after hydrothermal ageing was somewhat less dramatic to that of injection molded PLA, as reported by Yarahmadi et al. [24] under similar hydrothermal conditions. Ageing corresponding to 1 year of use significantly degraded PLA, so mechanical properties could not be measured. In addition, PLA ageing significantly decreased Tg, Tcc, and Tm.”
Conclusion, Line 304-305:
“After comparing the results of this study with those in previous studies on molded PLA, it can be concluded that FFF processed PLA had slower degradation kinetics.”
Point 2: The Introduction needs to be written in a manner that the flow is clear. As of now, it appears that the sentences are disjointed.
Response 2: The manuscript was undergone extensive English revisions by translator.
Point 3: Missing commas in multiple places. For example, Page1, Line 18. by 2030,
Response 3: The manuscript was undergone extensive English revisions by translator.
Point 4: Page 2, Line 44: Incomplete sentence
Response 4: Sentences in Row 44 were rewritten, Line 39-44.
“Some works tried to solve this problem. For example, Baechler et al. [18] developed an extruder to produce filament from polymer waste; Filabot [19]designed a plastic filament maker that uses post-consumer plastic; Woern et al. [20] designed an open source recycling system which costs less than $700 in materials and can be fabricated in about 24 h; and Dongoh Lee et al. [21] designed a recycling system to make post-consumer filaments for 3-D printers.”
Point 5: Page 2, Line 63-64: Reword sentence.
Response 5: Sentence were reworded, Line 64-65.
“Most studies have reported that hydrolytic degradation of PLA takes place in the bulk of the material rather than at the surface [31].”
Point 6: Page 3, Line 91: provide correct print temperature. Also indicate that a skirt was used. Were all 100 samples printed in one go or in batches?
Response 6: Paragraph 2.2 Printing has been changed accordingly in the manuscript, Line 89-99.
“A total of 90 regular dog-bone (type A2) specimens for thermal and hydrothermal ageing were printed in batches of five considering the ISO 20753 standard [32] and using a Witbox 2 printer (BQ) with operating temperatures of 200 ºC and a printing glass table without heater. Five specimens are used for each ageing cycle, and one extra specimen for contingency. All specimens were printed indoors in a temperature-controlled environment(23 ± 2 ºC and 50 ± 5 %R.H., as defined in [27]) with 100% infill, horizontal pattern orientation 0/90 (i.e., alternating layers with orientations at 0º and 90º), and layer height of 0.2 mm. Each specimen was printed with a skirt of two extruded perimeters that composed the outermost edge. The selected deposition pattern orientations and layer height offered better quasi-isotropic mechanical properties for PLA parts manufactured by FFF compared with 45/45 orientation [33,34].”
Point 7: Page 3, Line 109: It is not clear as to what the three points are.
Response 7: This sentence has been re-written to make it clearer, Line 114-115.
“Points were as follows: in the laboratory air, on one of the specimens surface, and in the central point of the oven air without touching any surface.”
Point 8: Page 4: Line 122-123. Multiple ISO standards are referenced in the manuscript without indicating what they stand for. I can be stated as for example, atmosphere defined in ISO 291 - Plastics - Standard atmospheres for conditioning and testing.
Response 8: The sentence has been changed accordingly in the manuscript, Line 124-128.
“Thickness, width and length of each specimen were measured three times before and after ageing with a micrometre, as per ISO 16012 - Plastics - Determination of linear dimensions of test specimens [44]. Speciments were conditioned for at least 88 h at 23 ± 2 ºC and 50 ± 5 % R.H. atmosphere, as defined in ISO 291 - Plastics - Standard atmospheres for conditioning and testing [27] with ±0.02 mm accuracy.”
Point 9: Page 4: Line 152 - Reword sentence. Not clear.
Response 9: The sentence was reworded, Line 153-155.
“An extensometer with a nominal length of 50 mm was used, but many specimens broke outside the gage length due to assumed stress concentrations in the regions where geometry changes as in [34].”
Point 10: Page 5: Line 178 - Table 2 and Fig. 1 convey same information. Use one of them.
Response 10: Table 2 has been suppressed in the amended manuscript.
Point 11: Page 5: Line 183: Numeral?
Response 11: Word “Numeral” has been suppressed in the amended manuscript, Line 185-187.
“Values of shrinkage ratios of thermal aged PLA had no constant tendency; however, there is a rising tendency in hydrothermal samples.”
Point 12: Page 5, Line 194: Reword sentence.
Response 12: Sentence were reworded, Line 194-195.
“Afterwards, sizes would stay the same for 1-2 years, with a maximum shrinkage ratio of 8% after water and temperature influences.”
Point 13: Page 7, Line 231: add Section before 3.4
Response 13: Word Section has been added before 3.4, Line 247-249.
“As crystallinity is an important characteristic affecting the mechanical properties [58] its influence on mechanical properties will be shown in Section 3.4.”
Point 14: Page 8, Line 245: have not must be have no
Response 14: Sentence has been rewritten, Line 264-265.
“Consequently, 3D printed PLA goods had no predictable thermal behaviour during their life cycle because of the unstable thermal nature of PLA.”
The numbering of Tables and Figures are incorrect in the version with tracking changes because of Latex program configurations. The right numbers are in the final version.

Reviewer 2 Report
The authors have prepared specimens via FFF-based 3D printing using PLA as polymeric material and exposed the printed samples to thermal and hydrothermal aging for various timeframes in the range of 0-1344 hours. Subsequently, they analyzed the dimensional stability, the thermal and mechanical properties of the fabricated specimens. The topic of paper could be of interest, however, the series of experiments does not seem to be well conducted and the English language also leaves something to be desired. The discussion and the representation of the data is simply poor. Below I list my most crucial remarks about the manuscript. Overall, I do not support this manuscript to be accepted in the Polymers journal.
The English language of the manuscript is generally poor, not to mention that the spelling of the words sometimes follows the American English, while in other cases the authors use British English. There are numerous unjustified words starting with capital letter (e. g. Polylactic acid; Additive Manufacturing). Abbreviations defined without being used later on (AM in the Abstract) are also present. There are also a high number of incomprehensible sentences and main clauses. Paragraphs of 1-2 sentences should also be avoided.
In some cases the referencing is inserted after the punctuation, other times before that. Sometimes there is a space before that other times there isn’t.
Spaces between values and units are also randomly used throughout the whole manuscript, also, the units are sometimes written in italic, other times they are not.
Rows 11-14 “This study reveals that … the crystallinity increases and tensile strength decreases. So, it can be assumed that PLA waste around two years old can be recycled without significant detriment of properties compared to virgin PLA.” – It is an obvious self-contradiction. How is decreasing strength no significant deterioration of properties?
Row 23 “The cumulative greenhouse gas emissions reduction was estimated at 92.8-217.4 Mt.” – When? In what timeframe?
Row 27-31 “Its relatively high tensile strength and modulus compared to other thermoplastics…make it inadequate for more demanding applications.” - How does high strength and modulus make it inadequate?
Row 44-45 “Filabot, a plastic filament maker that utilizes 44 post-consumer plastic[19] Woern et al. designed the open source recycling system which 45 costs less than $700 in materials and can be fabricated in about 24 h.” – I cannot even understand what the authors wish to describe in this sentence.
Row 86 “bending temperature under load -56°C” – firstly, please define what you mean by bending temperature under load; secondly, there is no transition of PLA at -56 °C justifying any changes in properties
In section 2.2. the authors state they used an optimum operating temperature of 150-250 °C for the 3D printing process. Using this wide range of temperatures for the different samples makes the whole experiment pointless, since the difference between the temperatures of the printing can influence the material’s structure crucially. Still in this section the authors claim they used a “cold” printing glass table. What exactly do you mean by cold? To what temperature was it cooled down to? Still in the same section the authors claim they have performed the printing in a temperature-controlled environment, yet, no information is provided about this temperature. Still in this section, according to the authors, they used an infill pattern orientation of 0/90, which they assume is optimal according to the literature. Assuming it was a linear pattern – which they never specified – the 0° is indeed the most optimal regarding the mechanical properties. On the other hand, the 90° infill generally results in the worst properties. There is no way to figure out, which one the authors actually used.
In Table 1, where the authors provide the correlations between aging time and service time, both the 16 h and 24 h of aging is stated to be equivalent with 7-14 days of ”Approximate time”. What was the reason for performing both of those if there is literally no difference according to the correlation?
In section 2.6. the authors state they used a heat-flux thermomechanical analyzer Q20. As far as I know there are no thermomechanical analyzers working on heat-flux principle.
In Table 2 the authors summarize the dimensional values measured on the specimens after the specific aging times and the calculated shrinkage ratios. In here, I have multiple remarks. Firstly, the initial values of the specimens are not reported, only the aged samples. Secondly, providing the individual values is rather unnecessary, since they are not evaluated within the discussion at all. As such, they are just making Table 2 “busy” and difficult to read. Reporting the shrinkage would be sufficient.
Figure 1 provides no additional information compared to Table 2, it is simply a duplication of the data. Still here, no standard deviation of the data is provided, as such it is quite irresponsible to make any statements about trends within the discussion, as the authors did.
In row 198-199 the authors provide information regarding the Tc of pure PLA, where it “gives off heat”. Until this point the authors never mentioned using any other materials than pure PLA.
In Figure 2 the two diagrams should have the same Y scale. In this current form they are simply not comparable.
In row 202-205 the authors conclude that the aged materials are stiffer, due to having higher Tg. How is higher Tg related with greater stiffness?
In Table 3 it is not clear which melting temperature of PLA the authors considered as Tm. As it can be seen in multiple curves, PLA exhibits a double melting behavior, which is also not explained.
According the information provided in Table 3 the enthalpy of crystallization in many cases exceeds the enthalpy of melting. In the most extreme case – 48 hours of thermal aging – almost by 20% (32 J/g vs 27 J/g) How is that even possible? The crystalline structure still exists after the melting phenomenon?
Row 223: What do you mean by “infinitely large crystal”?
Row 224-225 For hydrothermally aged samples the authors concluded that the crystallinity increased from 19% to 42%, that is, however, only true between 24 h and 1344 h of aging. Meanwhile, there is a considerable, 9% of decrease between 0 and 24 h that they never explained.
In row 237 the authors discuss the thermal properties of virgin PLA. Until this point the authors never mentioned using any other materials than virgin PLA.
In rows 238-241 there is a redundancy. In two consecutive sentences the authors conclude twice that PLA is pure without any contamination.
In rows 241-242 the authors state there is only a single degradation peak for PLA. There is, however, no peak to be observed in the diagrams of Figure 3.
In rows 243-246 the authors discuss the changes in the Tloss5% temperatures of the various samples, however, these values were never reported in the manuscript.
In rows 253-257 the authors attribute the increasing strength of the PLA during the first several hours of thermal aging to the evaporation of the water. Where is this water coming from? How did it not evaporate during the 3D printing process? How come the modulus did not increase if the water really did evaporate?
Author Response
Point 1: The English language of the manuscript is generally poor, not to mention that the spelling of the words sometimes follows the American English, while in other cases the authors use British English. There are numerous unjustified words starting with capital letter (e. g. Polylactic acid; Additive Manufacturing). Abbreviations defined without being used later on (AM in the Abstract) are also present. There are also a high number of incomprehensible sentences and main clauses. Paragraphs of 1-2 sentences should also be avoided.
Response 1: The manuscript was undergone extensive English revisions by translator. Unused abbreviations have been deleted.
Point 2: In some cases the referencing is inserted after the punctuation, other times before that. Sometimes there is a space before that other times there isn’t.
Response 2: Referencing has been placed inside the punctuation according to the Polymers guide for authors.
Point 3: Spaces between values and units are also randomly used throughout the whole manuscript, also, the units are sometimes written in italic, other times they are not.
Response 3: The use of spaces has been adapted following the conventions of the ‘Bureau international des poids et mesures’. From the SI Brochure, §5.3.3:
The numerical value always precedes the unit, and a space is always used to separate the unit from the number. (…) The only exceptions to this rule are for the unit symbols for degree, minute, and second for plane angle, °, ′, and ″, respectively, for which no space is left between the numerical value and the unit symbol.
The use of italics or roman fonts has been adapted in the amended manuscript to apply the IUPAC guidelines:
https://iupac.org/wp-content/uploads/2016/01/ICTNS-On-the-use-of-italic-and-roman-fonts-for-symbols-in-scientific-text.pdf
Point 4: Rows 11-14 “This study reveals that … the crystallinity increases and tensile strength decreases. So, it can be assumed that PLA waste around two years old can be recycled without significant detriment of properties compared to virgin PLA.” – It is an obvious self-contradiction. How is decreasing strength no significant deterioration of properties?
Response 4: The last sentence of the abstract has been changed to avoid self-contradiction, Row 10-12.
“This study revealed that tensile strength decreased after 1344 h of hydrothermal ageing, simulating 1.5-2.5 years of real service time. PLA waste mixture had therefore the same thermo-mechanical properties before reaching 1.5-year-old, so it could be recycled.”
Point 5: Row 23 “The cumulative greenhouse gas emissions reduction was estimated at 92.8-217.4 Mt.” – When? In what timeframe?
Response 5: Timeframe and more details on the scenario given in Ref. [2] have been included in the amended text, Row 20-22.
“From 2019 to 2050, the cumulative greenhouse gas emissions reduction of lightweight metallic aircraft components produced using AM is estimated at 92.8–217.4 Mt [2]”
Point 6: Row 27-31 “Its relatively high tensile strength and modulus compared to other thermoplastics…make it inadequate for more demanding applications.” - How does high strength and modulus make it inadequate?
Response 6: The word “besides” has been included in the beginning of the sentence, Row 25-29.
“Besides its relatively high tensile strength and modulus compared to other thermoplastics (e.g., polyethylene terephthalate and polypropylene), its low impact strength, less heat tolerance, brittleness with less than 10% elongation at break [4], low crystallization rate, and poor ductility make it inappropriate for more demanding applications [4,5].”
Point 7: Row 44-45 “Filabot, a plastic filament maker that utilizes 44 post-consumer plastic[19] Woern et al. designed the open source recycling system which 45 costs less than $700 in materials and can be fabricated in about 24 h.” – I cannot even understand what the authors wish to describe in this sentence.
Response 7: Sentences in Row 44-45 were rewritten, Row 40-44.
“Some works tried to solve this problem. Baechler et al. [18] developed an extruder to produce filament from polymer waste; McNaney [19] designed a plastic filament maker that uses post-consumer plastic; Woern et al. [20] designed an open-source recycling system which costs less than $700 in materials and can be fabricated in about 24 h; and Dongoh Lee et al. [21] designed a recycling system to make post-consumer filaments for 3-D printers.”
Point 8: Row 86 “bending temperature under load -56°C” – firstly, please define what you mean by bending temperature under load; secondly, there is no transition of PLA at -56 °C justifying any changes in properties
Response 8: The original draft mistakenly had the minus sign before some numbers of 2.1 section. The amended manuscript has corrected these typos. Full reference of the standards mentioned in this section have also been included, Row 83-87.
“Commercially available PLA filament of 1.75 mm width was purchased from BQ (Madrid, Spain), with a printing temperature range = 200-220 ºC, an optimum printing temperature = 205 ºC, a bending temperature under load = 56 ºC (ISO 75/2B), a melting temperature = 145-160 ºC (ASTM D3418), and a glass transition temperature = 56-64 ºC (ASTM D3418), as indicated in the product datasheet.”
Point 9: In section 2.2. the authors state they used an optimum operating temperature of 150-250 °C for the 3D printing process. Using this wide range of temperatures for the different samples makes the whole experiment pointless, since the difference between the temperatures of the printing can influence the material’s structure crucially. Still in this section the authors claim they used a “cold” printing glass table. What exactly do you mean by cold? To what temperature was it cooled down to? Still in the same section the authors claim they have performed the printing in a temperature-controlled environment, yet, no information is provided about this temperature. Still in this section, according to the authors, they used an infill pattern orientation of 0/90, which they assume is optimal according to the literature. Assuming it was a linear pattern – which they never specified – the 0° is indeed the most optimal regarding the mechanical properties. On the other hand, the 90° infill generally results in the worst properties. There is no way to figure out, which one the authors actually used.
Response 9: 150-250°C is the operating temperature range of the used 3D printer. All samples were printed at 200°C. “Cold” printing glass table means that 3D printer used doesn’t have option of heating platform to a selected temperature. Information about environment temperature is provided in the amended text according with reviewer comment. Information about pattern orientation is completed.
Section 2.2 has been corrected accordingly, Row 89-99.
“A total of 90 regular dog-bone (type A2) specimens for thermal and hydrothermal ageing were printed in batches of five considering the ISO 20753 standard [32] and using a Witbox 2 printer (BQ) with operating temperatures of 200 °C and a printing glass table without heater. Five specimens were used for each ageing cycle, and one extra specimen for contingency. All specimens were printed indoors in a temperature-controlled environment (23 ± 2 °C and 50 ± 5%R.H., as defined in [27]) with 100% infill, horizontal pattern orientation 0/90 (i.e., alternating layers with orientations at 0º and 90º), and layer height of 0.2 mm. Each specimen was printed with a skirt of two extruded perimeters that composed the outermost edge. The selected deposition pattern orientations and layer height offered better quasi-isotropic mechanical properties for PLA parts manufactured by FFF compared with 45/45 orientation [33–34].”
Point 10: In Table 1, where the authors provide the correlations between aging time and service time, both the 16 h and 24 h of aging is stated to be equivalent with 7-14 days of ”Approximate time”. What was the reason for performing both of those if there is literally no difference according to the correlation?
Response 10: We agree with the reviewer that this point could cause a misunderstanding. In the first version approximation was done to 1 week. As thermal ageing test was conducted in accordance with ISO 188 [1], results of both 16h and 24h time intervals were presented. To avoid misunderstanding approximate time ranges have been rounded up to days in the amended version.
Table 1. Correlation between ageing and service time following the Simplified Protocol for Accelerated Ageing.
|
Ageing time at 50±2°C |
Approximate time |
|
h |
days |
|
0 |
0 |
|
8 |
2.5-5 |
|
16 |
5-10 |
|
24 |
8-16 |
|
48 |
16-24 |
|
72 |
24-56 |
|
168 |
56-112 |
|
672 |
224-448 |
|
1344 |
448-896 |
Point 11: In section 2.6. the authors state they used a heat-flux thermomechanical analyzer Q20. As far as I know there are no thermomechanical analyzers working on heat-flux principle.
Response 11: The text has been amended accordingly to avoid misinterpretation, Row 130-132.
“A Q20 differential scanning calorimeter (DSC) from TA Instruments Inc. (USA) was used to measure the thermal properties of potential PLA waste, as indicated in ISO 11357-1 [45].”
Point 12: In Table 2 the authors summarize the dimensional values measured on the specimens after the specific aging times and the calculated shrinkage ratios. In here, I have multiple remarks. Firstly, the initial values of the specimens are not reported, only the aged samples. Secondly, providing the individual values is rather unnecessary, since they are not evaluated within the discussion at all. As such, they are just making Table 2 “busy” and difficult to read. Reporting the shrinkage would be sufficient.
Response 12: Following suggestions from Reviewer 1 and Reviewer 2, Table 2 has been deleted and only shrinkage information is shown graphically in Figure 1.
Point 13: Figure 1 provides no additional information compared to Table 2, it is simply a duplication of the data. Still here, no standard deviation of the data is provided, as such it is quite irresponsible to make any statements about trends within the discussion, as the authors did.
Response 13: Table 2 has been suppressed in the amended manuscript. Standard deviation was added to the Figure 1 as error bars.
Text has been included in Figure 1 caption: “Error bars are inserted considering standard deviations for six calculated shrinkage ratios.”
Point 14: In row 198-199 the authors provide information regarding the Tc of pure PLA, where it “gives off heat”. Until this point the authors never mentioned using any other materials than pure PLA.
Response 14: The text has been amended accordingly to avoid misinterpretation, Row 200-201.
“Unaged PLA reached this point at 117 ◦C. The third peak showed that the polymer achieved its melting point (Tm = 151 ◦C for unaged PLA).”
Point 15: In Figure 2 the two diagrams should have the same Y scale. In this current form they are simply not comparable.
Response 15: In Figure 2 the heat flow is plotted on Y-axis. Firstly, Y scale was not shown because the heat flux difference was not discussed in the text. Only difference in glass transition and melting temperatures was compared. Secondly, representation of the numerical values of the heat flow on the graph makes it difficult to distinguish the temperatures’ difference. Thirdly, most authors use this type of Figure, for example [2–5].
The explanation has been included in Figure 2 caption: “The graphs are shifted in the Y-axis to better compare peak temperatures.”
Point 16: In row 202-205 the authors conclude that the aged materials are stiffer, due to having higher Tg. How is higher Tg related with greater stiffness?
Response 16: We rewrote this sentence in order to make it clear, Row 217-217.
“Nevertheless, the results show that the Tg of aged PLA is c11equal or greater than that measured for unaged sample, so the aged material c12requires greater temperature to become rubber-like.”
Point 17: In Table 3 it is not clear which melting temperature of PLA the authors considered as Tm. As it can be seen in multiple curves, PLA exhibits a double melting behavior, which is also not explained.
Response 17: Row 222-224 explain this phenomenon: “DSC thermograms of thermal and hydrothermal ageing have two melting peaks after 672 and 1344 h of treatment. The reason could be the annealing during the DSC scans, where some regions of the material recrystallize and remelt [8].“ Therefore, the maximum peak is considered as Tm. Nevertheless, a comment has been introduced in Table 2 caption to make it clear:
“Table 2. Results of the thermal analysis of specimens after ageing. Samples which registered a double peak when melting are marked with an asterisk (*). The second peak (the one at greater temperature) is also shown, considered as Tm.”
Point 18: According the information provided in Table 3 the enthalpy of crystallization in many cases exceeds the enthalpy of melting. In the most extreme case – 48 hours of thermal aging – almost by 20% (32 J/g vs 27 J/g) How is that even possible? The crystalline structure still exists after the melting phenomenon?
Response 18: According to the received data, enthalpies of crystallization exceed the enthalpies of melting for thermally aged samples during 16, 24, 48, 72 h. The relation ∆Hc≈∆Hm indicates that almost the entire crystal phase in PLA forms during the cold crystallisation, which excludes the crystallisation from the liquid phase [6]. As printed samples were cooled down at ambient temperature it was resulted in quenching of the molten PLA [7]. The inequality ∆Hc >∆Hm, observed for thermally aged samples during 16, 24, 48, 72 h, may be the result of an indistinct separation of the processes of crystallisation and melting, which leads to a partial superposition of the relevant peaks. The same phenomena was observed by [6]. The text has been amended accordingly to avoid misunderstandings, Row 229-237.
Point 19: Row 223: What do you mean by “infinitely large crystal”?
Response 19: This is direct reference from [8]. Also, instead “infinitely large crystal”, it can be said “100% crystalline PLA” [9]
Point 20: Row 224-225 For hydrothermally aged samples the authors concluded that the crystallinity increased from 19% to 42%, that is, however, only true between 24 h and 1344 h of aging. Meanwhile, there is a considerable, 9% of decrease between 0 and 24 h that they never explained.
Response 20: There are several methods to count the degree of crystallinity. First time, we used method presented in [5]. As it is happening in [10] (difference between PET and PE) it seems that for Tg well bellow ambient temperature the method for calculating Xc that we used is good, but for polymers which Tg is over ambient temperature the calculation gives a very large value.
So we recalculated the degree of crystallinity according to [11], considering ∆Hc, as
Xc=(∆Hm-∆Hc)/∆H*x100
Received data confirm the statement, that the crystallinity increased, and it is true between 0 h and 1344 h of ageing. Meanwhile, there is no decrease between 0 and 24 h. So, recalculated data were included in Table 2. The text and value of crystallinity have been amended accordingly.
|
|
Ageing time |
Tg |
Tc |
∆Hc |
Tm |
∆Hm |
Xc |
Tloss5% |
|
h |
ºC |
ºC |
J/g |
ºC |
J/g |
% |
°C |
|
|
Unaged |
0 |
58 |
118 |
27 |
151 |
28 |
1.1 |
292 |
|
Thermal ageing |
8 |
61 |
118 |
27 |
153 |
28 |
1.1 |
309 |
|
16 |
61 |
118 |
30 |
153 |
28 |
- |
301 |
|
|
24 |
60 |
118 |
29 |
153 |
28 |
- |
281 |
|
|
48 |
59 |
117 |
32 |
149 |
27 |
- |
294 |
|
|
72 |
60 |
115 |
32 |
149 |
29 |
- |
272 |
|
|
168 |
62 |
115 |
26 |
150 |
28 |
2.2 |
298 |
|
|
672 |
59 |
113 |
29 |
149 |
29 |
0 |
303 |
|
|
1344 |
59 |
112 |
27 |
149 |
28 |
1.1 |
266 |
|
|
Hydrothermal ageing |
24 |
58 |
125 |
15 |
151 |
18 |
3.2 |
311 |
|
48 |
59 |
123 |
16 |
149 |
20 |
4.3 |
305 |
|
|
72 |
58 |
123 |
16 |
153 |
20 |
4.3 |
278 |
|
|
168 |
57 |
115 |
23 |
149 |
27 |
4.3 |
287 |
|
|
672 |
59 |
111 |
27 |
147 |
31 |
4.3 |
303 |
|
|
1344 |
58 |
104 |
31 |
155 |
39 |
8.6 |
264 |
Point 21: In row 237 the authors discuss the thermal properties of virgin PLA. Until this point the authors never mentioned using any other materials than virgin PLA.
Response 21: ‘Unaged’ has been used instead of ‘virgin’., Row 257.
“Unaged PLA was very stable up to 300 ◦C.”
Point 22: In rows 238-241 there is a redundancy. In two consecutive sentences the authors conclude twice that PLA is pure without any contamination.
Response 22: We rewrote this sentences in order to avoid repetition, Row 257-259.
Point 23: In rows 241-242 the authors state there is only a single degradation peak for PLA. There is, however, no peak to be observed in the diagrams of Figure 3.
Response 23: The text has been amended accordingly, Row 259-260.
“Additionally, all samples showed one single weight reduction step, indicating the presence of one type of polymer (PLA).”
Point 24: In rows 243-246 the authors discuss the changes in the Tloss5% temperatures of the various samples, however, these values were never reported in the manuscript.
Response 24: The text has been amended accordingly, Row 261.
“Values of Tloss5% (last column in Table 3) indicated that thermal stability of aged PLA did not constantly increase or decrease.”
Point 25: In rows 253-257 the authors attribute the increasing strength of the PLA during the first several hours of thermal aging to the evaporation of the water. Where is this water coming from? How did it not evaporate during the 3D printing process? How come the modulus did not increase if the water really did evaporate?
Response 25: PLA is hygroscopic material, so it absorbs water from the air. In our experiment, PLA could first absorb water during the time between printing and ageing. Second, water absorption might occur between ageing and tensile test during conditioning of the samples for more than 88h at 23±2 ◦C and 50±5%R.H. according to ISO 527-2. In the manuscript, we suggested that unaged PLA samples saved their relatively high water absorption ability in comparison with aged samples because of molecular movements that were discussed in Section 3.1. This statement was presented as a suggestion and needed further study. The text has been amended accordingly, Row 273-276.
“This value is greater than the value of non-aged polymer. PLA is hygroscopic, it absorbs water from ambient humidity which may subsequently be evaporated in the first 8 h of thermal ageing [31]. Water may disappear between PLA molecules, so they become closer to each other, thus improving tensile strength.”
Reference
- ISO/TC 45/SC 2 Testing and analysis ISO 188:2011 Rubber, Vulcanized or Thermoplastic — Accelerated Ageing and Heat Resistance Tests 2013, 20.
- Yarahmadi, N.; Jakubowicz, I.; Enebro, J. Polylactic Acid and Its Blends with Petroleum-Based Resins: Effects of Reprocessing and Recycling on Properties. J. Appl. Polym. Sci. 2016, 133, 1–9, doi:10.1002/app.43916.
- Beltrán, F.R.; Barrio, I.; Lorenzo, V.; del Río, B.; Martínez Urreaga, J.; de la Orden, M.U. Valorization of Poly(Lactic Acid) Wastes via Mechanical Recycling: Improvement of the Properties of the Recycled Polymer. Waste Manag. Res. 2019, 37, 135–141, doi:10.1177/0734242X18798448.
- Beltran, F.R.; Arrieta, M.P.; Diego, E.; Lozano-Perez, A.A.; Cenis, J.L.; Gaspar, G.; U de la Orden, M.; Urreaga, J.M. Effect of Yerba Mate and Silk Fibroin Nanoparticles on the Migration Properties in Ethanolic Food Simulants and Composting Disintegrability of Recycled PLA Nanocomposites. Polymers (Basel). 2021, 13, 1925, doi:https://doi.org/10.3390/ polym13121925.
- Beltrán, F.R.; Lorenzo, V.; Acosta, J.; de la Orden, M.U.; Martínez Urreaga, J. Effect of Simulated Mechanical Recycling Processes on the Structure and Properties of Poly(Lactic Acid). J. Environ. Manage. 2018, 216, 25–31, doi:10.1016/j.jenvman.2017.05.020.
- Zenkiewicz, M.; Richert, J.; Rytlewski, P.; Moraczewski, K.; Stepczyńska, M.; Karasiewicz, T. Characterisation of Multi-Extruded Poly(Lactic Acid). Polym. Test. 2009, 28, 412–418, doi:10.1016/j.polymertesting.2009.01.012.
- Pluta, M.; Galeski, A.; Alexandre, M.; Paul, M.A.; Dubois, P. Polylactide/Montmorillonite Nanocomposites and Microcomposites Prepared by Melt Blending: Structure and Some Physical Properties. J. Appl. Polym. Sci. 2002, 86, 1497–1506, doi:10.1002/app.11309.
- Pillin, I.; Montrelay, N.; Bourmaud, A.; Grohens, Y. Effect of Thermo-Mechanical Cycles on the Physico-Chemical Properties of Poly(Lactic Acid). Polym. Degrad. Stab. 2008, 93, 321–328, doi:10.1016/j.polymdegradstab.2007.12.005.
- Aouat, T.; Kaci, M.; Lopez-Cuesta, J.M.; Devaux, E. Investigation on the Durability of PLA Bionanocomposite Fibers Under Hygrothermal Conditions. Front. Mater. 2019, 6, 1–15, doi:10.3389/fmats.2019.00323.
- Sarge, S.M.; H, W.H.; Hemminger, W. Calorimetry_ Fundamentals, Instrumentation and Applications-Wiley-VCH (2014); ISBN 9780813814834.
- Nam, P.H.; Maiti, P.; Okamoto, M.; Kotaka, T.; Hasegawa, N.; Usuki, A. A Hierarchical Structure and Properties of Intercalated Polypropylene/Clay Nanocomposites. Polymer (Guildf). 2001, 42, 9633–9640, doi:10.1016/S0032-3861(01)00512-2.
- Gorrasi, G.; Pantani, R. Hydrolysis and Biodegradation of Poly ( Lactic Acid ). Adv. Polym. Sci. 2018, 279, 119–152, doi:10.1007/12.
The numbering of Tables and Figures are incorrect in the version with tracking changes because of Latex program configurations. The right numbers are in the final version.

Reviewer 3 Report
Characterization of mechanical properties - What was the repeatability of the individual measurements? Please add repeatability to the text.
Author Response
Point 1: Characterization of mechanical properties - What was the repeatability of the individual measurements? Please add repeatability to the text.
Response 1: Repeatability of the individual measurements has been included to Table 3.
The following text has been embedded accordingly to Section 2.8, Line 170-172.
“The repeatability of the individual measurements is calculated as the difference between the maximum and minimum values among 5 samples.”
Table 3. Tensile properties of 3D printed specimens after ageing.
|
|
Ageing time |
Tensile Yield Strength |
Young’s Modulus |
Elongation |
||||||
|
Average |
Standard deviation |
Repeat-ability |
Average |
Standard deviation |
Repeat-ability |
Average |
Standard deviation |
Repeat-ability |
||
|
h |
MPa |
MPa |
MPa |
MPa |
MPa |
MPa |
% |
% |
MPa |
|
|
Unaged |
0 |
28.746 |
1.002 |
2.226 |
1800.208 |
150.395 |
323.839 |
2.34 |
0.19 |
0.45 |
|
Thermal ageing |
8 |
31.189 |
1.322 |
3.378 |
2158.489 |
117.927 |
307.523 |
2.19 |
0.14 |
0.33 |
|
16 |
33.164 |
1.129 |
2.775 |
2115.412 |
157.063 |
356.287 |
2.39 |
0.16 |
0.37 |
|
|
24 |
36.799 |
1.154 |
2.894 |
2134.546 |
286.843 |
475.852 |
2.53 |
0.21 |
0.50 |
|
|
48 |
29.135 |
1.681 |
3.845 |
1612.385 |
312.078 |
429.898 |
2.49 |
0.22 |
0.54 |
|
|
72 |
29.721 |
1.551 |
3.875 |
1875.389 |
71.578 |
152.065 |
2.28 |
0.21 |
0.56 |
|
|
168 |
29.516 |
0.495 |
1.133 |
1968.879 |
102.064 |
261.783 |
2.05 |
0.23 |
0.57 |
|
|
672 |
32.602 |
0.477 |
0.927 |
2141.455 |
48.957 |
123.278 |
2.06 |
0.46 |
0.98 |
|
|
1344 |
31.126 |
2.851 |
7.579 |
2077.937 |
186.864 |
443.046 |
2.00 |
0.46 |
0.66 |
|
|
Hydrothermal ageing |
24 |
33.454 |
1.238 |
3.328 |
1552.181 |
74.724 |
177.765 |
3.16 |
0.15 |
1.30 |
|
48 |
32.672 |
2.040 |
4.950 |
1652.115 |
243.715 |
651.275 |
2.94 |
0.43 |
1.06 |
|
|
72 |
32.530 |
1.218 |
2.943 |
1340.578 |
322.173 |
825.923 |
3.05 |
0.29 |
0.60 |
|
|
168 |
29.613 |
1.259 |
2.441 |
1664.317 |
144.847 |
365.233 |
2.81 |
0.49 |
1.20 |
|
|
672 |
29.816 |
2.190 |
4.598 |
1962.120 |
94.488 |
237.910 |
2.40 |
0.57 |
1.35 |
|
|
1344 |
20.890 |
4.240 |
10.727 |
1598.108 |
383.587 |
936.971 |
1.74 |
0.19 |
0.46 |
|
The numbering of Tables and Figures are incorrect in the version with tracking changes because of Latex program configurations. The right numbers are in the final version.

Round 2
Reviewer 2 Report
The authors performed a great deal of corrections. The text was also obviously checked by a language polishing expert for general corrections, however, the field specific terms are still very poorly used, sometimes the text is misleading. The comments I can still not agree with:
Point 6. „Besides” is not the proper word to use in here. I would suggest „Despite” instead.
Point 25. In here I still cannot agree with the answers of the authors. If the material really absorbed some water after the printing and before the testing, then it should be mentioned within the text, that the 0 hour test was not performed right after the specimen coming out of the 3D printer, and the samples had enough time to absorb the environmental humidity. Even with accepting the fact above, the explanations provided by the authors in the text is also rather doubtful considering that the strength of samples thermally aged for 48 hours being almost the same as the strength of unaged samples (way within deviation range). The reported strength after 1344 h of aging is also within deviation range with the unaged sample, as such it is reckless to state that it has grown, unless a statistical significance test was also performed. Seeing all the strength data, I suggest performing the statistical analysis, which will most probably make it clear that there was no significant change in the strength over the whole thermal aging process.
Also, what do the authors mean by “water may disappear between PLA molecules”?
Author Response
Response to Reviewer 2 Comments
The authors performed a great deal of corrections. The text was also obviously checked by a language polishing expert for general corrections, however, the field specific terms are still very poorly used, sometimes the text is misleading. The comments I can still not agree with:
Point 6. „Besides” is not the proper word to use in here. I would suggest „Despite” instead.
Response 6. The word “despite” has been used instead of „besides” in the beginning of the sentence in row 25.
Point 25. In here I still cannot agree with the answers of the authors. If the material really absorbed some water after the printing and before the testing, then it should be mentioned within the text, that the 0 hour test was not performed right after the specimen coming out of the 3D printer, and the samples had enough time to absorb the environmental humidity. Even with accepting the fact above, the explanations provided by the authors in the text is also rather doubtful considering that the strength of samples thermally aged for 48 hours being almost the same as the strength of unaged samples (way within deviation range). The reported strength after 1344 h of aging is also within deviation range with the unaged sample, as such it is reckless to state that it has grown, unless a statistical significance test was also performed. Seeing all the strength data, I suggest performing the statistical analysis, which will most probably make it clear that there was no significant change in the strength over the whole thermal aging process.
Also, what do the authors mean by “water may disappear between PLA molecules”?
Response 25. It has been mentioned in Section 2.8, that tensile test of all samples, unaged and aged, was conducted after conditioning. Row 149-150
“All tests were carried out after dimensional measurements, so specimens were already conditioned (see section 2.5).”
According to reviewer’s suggestion, an analysis of variance (ANOVA) has been performed. The paper text has been amended accordingly. Row 274-290.
“The results of tensile testing with 95% confidence intervals of values, according to the procedure given in ISO 2602 [59], are shown in Table 3 and Figure 4. An analysis of variance (ANOVA) has been performed using StatGraphics software (Statgraphics Technologies, Inc., USA) to check whether the differences observed among the tensile strength mean values are statistically significant. Since the calculated p-value of the ANOVA F-test is zero (less than 0.05), there is a statistically significant difference between the yield strength from one thermal ageing time to another at a confidence level of 95.0 %. Therefore, it can be stated that, for thermally aged samples, tensile strength rises during the first 24 h. Then, after 48 h, tensile strength reduces and does not show significant changes after further ageing. It is important to note that all specimens for tensile analysis have been tested after determining geometric sizes, and, following ISO 291, they were conditioned for 88 hours with standard temperature and relative humidity. As PLA is hygroscopic, after 3D printing, specimens could reabsorb water molecules from ambient humidity which are weakly bound to the carbonyl oxygen atom of PLA through dipole interactions. Water may subsequently be evaporated in the first 24 h of thermal ageing, so the PLA chains are able to create more Van Der Waals interactions between them, thus improving tensile strength. The reduction in strength observed after 48 h of thermal ageing is consistent with the slight decrease in Tc and Tm mentioned in section 3.2, which suggests a chain length shortening.”

Round 3
Reviewer 2 Report
The authors made a lot of effort for their paper to be published.